# Video World Models with Long-term Spatial Memory

**Tong Wu*[1], Shuai Yang*[2,4], Ryan Po[1], Yinghao Xu[1],**
**Ziwei Liu[5], Dahua Lin[3,4], Gordon Wetzstein[1]**
[1] Stanford University    [2] Shanghai Jiao Tong University    [3] The Chinese University of Hong Kong
[4] Shanghai Artificial Intelligence Laboratory    [5] S-Lab, Nanyang Technological University

`https://spmem.github.io/`

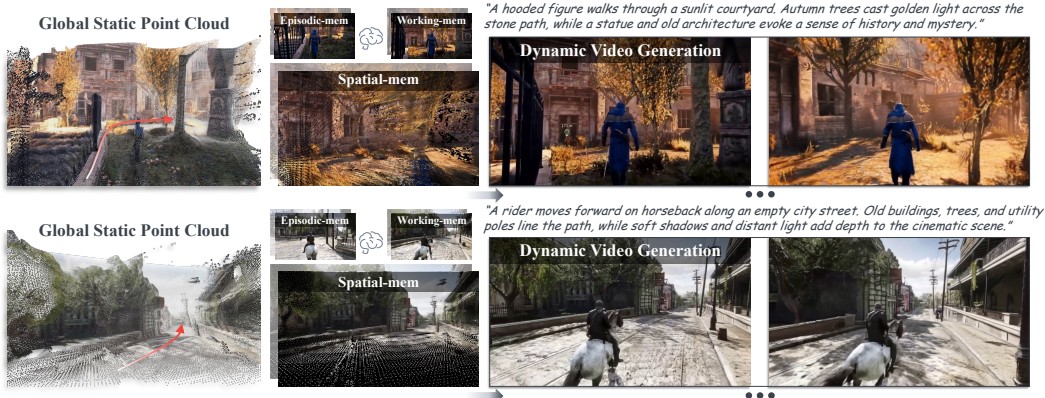

Figure 1: We augment video world models with memory. In this context, we consider the conventional approach of conditioning autoregressively generated frames with a few recent context frames as a short-term working memory. We explore two additional mechanisms modeling different types of long-term memory: spatial and episodic memory. The former is represented as a point map that is autoregressively generated along with the video frames and fused into the spatial memory by extracting only its static scene parts. To remember visual detail and identities for long time horizons, we also store a sparse set of historical reference frames as an episodic memory. Together, our memory mechanisms significantly improve the long-term consistency of emerging video world models.

## Abstract

Emerging world models autoregressively generate video frames in response to actions, such as camera movements and text prompts, among other control signals. Due to limited temporal context window sizes, these models often struggle to maintain scene consistency during revisits, leading to severe forgetting of previously generated environments. Inspired by the mechanisms of human memory, we introduce a novel framework to enhancing long-term consistency of video world models through a geometry-grounded long-term spatial memory. Our framework includes mechanisms to store and retrieve information from the long-term spatial memory and we curate custom datasets to train and evaluate world models with explicitly stored 3D memory mechanisms. Our evaluations show improved quality, consistency, and context length compared to relevant baselines, paving the way towards long-term consistent world generation.

39th Conference on Neural Information Processing Systems (NeurIPS 2025).

# 1 Introduction

World models are generative systems that learn to predict an environment in response to actions, making them well suited for simulating complex, interactive settings [28, 2, 30, 74, 90]. Video diffusion models [11, 37, 44, 79, 55] have emerged as a powerful approach to architecting world models, especially when used with autoregressive next-frame prediction [1, 12, 18, 22, 41, 53, 60, 65, 73, 81, 35]. Existing video generation models, however, often struggle with long-horizon consistency due to limited temporal context windows, frequently forgetting previously seen scenes during revisits. This is due to the relatively small number of previously generated context frames that the model can consider when generating new frames—a problem primarily caused by the quadratic growth of computational complexity in the attention module of the underlying diffusion transformers.

To address this challenge, current world models simply keep the number of context frames low to maintain computational feasibility. Several very recent approaches explore progressive downsampling of temporally more distant frames to increase the temporal context window size [25, 86]. Yet, all these approaches rely on image-based representations of the past and lack a persistent 3D understanding of the world, limiting spatial consistency.

Inspired by the mechanisms of human memory [3], we propose a new framework to enhance the long-term consistency of video world models through long-term spatial memory grounded in geometry. Drawing from cognitive theories, our approach incorporates three distinct forms of memory—***spatial, working, and episodic***—each modeled through a dedicated representation. Similar to existing models, our framework relies on a set of recently generated context frames. We consider this a *short-term working memory* mechanism, making both static and dynamic aspects of the most recent past accessible in the form of pixel data. To help remember long-term spatial relationships, we introduce an additional *long-term spatial memory*. This mechanism primarily relies on an explicit 3D representation of the generated world, augmented by a *sparse episodic memory* in the form of a set of keyframes from the past. We implement the spatial memory using a geometry-grounded point cloud representation. Before storing newly generated information into this memory mechanism, we filter out dynamic parts of the world to primarily remember the static parts of the work in this memory.

The primary contribution of our work is the design of our memory mechanism, combining short-term working memory with long-term spatial and sparse episodic memory, as illustrated in Figure 1. We develop approaches to store newly generated information into the spatial and episodic memory bank as well as retrieve information from it to effectively condition the generation of new video frames. Moreover, we curate a custom dataset to train and evaluate a proof-of-principle implementation of the proposed mechanisms. Our evaluations show that the quality and 3D consistency of our approach surpasses that of relevant baselines. With this work, we hope to contribute to the growing community effort of unlocking infinite-length, consistent world generation capabilities for computer graphics, robotics, and other interactive applications.

# 2 Related work

We consider a world model to be a generative system that autoregressively generates image or video frames, conditioned on actions or camera pose. This work builds on several important concepts in the generative AI literature, which we briefly review in the following.

**Image and video generation.**     In recent years, diffusion models have emerged as the state-of-the-art paradigm for image generation [31], surpassing the performance of GAN-based methods in both fidelity and diversity by modeling complex distributions through iterative denoising [45, 51, 64, 67, 20]. These successes have naturally extended to the task of video generation, as diffusion-based architectures have been adapted to include the temporal domain, enabling the generation of high-quality video clips spanning the order of tens to hundreds of frames [89, 48, 78, 56, 21, 14, 8]. Additional advancements in efficiency and controllability have also followed, through techniques such as improved tokenization [63, 75] and flow-matching objectives [49, 40].

**Autoregressive video generation.**     To support generation of longer videos with online inference capabilities, autoregressive (AR) approaches and architectures have been introduced in the diffusion framework [62, 43, 76, 71]. The autoregressive regime is also particularly intuitive, given the natural temporal ordering of video data. Taking inspiration from LLMs, state-of-the-art methods model

video data by processing frames into spatio-temporal tokens [48, 89, 78, 63, 21, 56, 14], learning to autoregressively generate new tokens until a full video is formed. This is in stark contrast with conventional diffusion-based frameworks, which choose to iteratively denoise an entire image/video at once [9, 46, 37, 58, 79, 44, 29, 52]. Recent approaches include training conditional diffusion models that generate new frames given a set of previous clean frames, allowing AR generation simply by passing newly generated frames as context for future frames [37, 34, 16, 41, 88, 24, 23]. Others modify the diffusion objective by assigning independent noise levels for each frame during training, which also supports AR inference [13, 80, 60]. In practice, these methods can be used for infinite-length generations by utilizing a sliding window context, but suffers from limited memory and drift.

**Controlled video generation.** Controlled or conditional video generation is a core component of world simulators. A series of works have explored explicit camera control, including [72, 32, 33, 33, 5, 4], which enable novel-view or multi-view generation by disentangling camera trajectories from dynamic content. Most of these approaches take either a single image or a text prompt as input, and often struggle to maintain long-term consistency, especially in dynamic scenes. More recent efforts [85, 7, 6] focus on re-filming or generating synchronized views in dynamic settings, further advancing controllability. Beyond direct camera injection, structural conditioning via point clouds, tracking, or 3D-aware priors has proven effective for improving spatial consistency and trajectory alignment [83, 57, 26, 82, 15]. In addition to spatial and motion-level control, some models support action-based or scene-level conditioning [54, 19], where either structured action vocabularies or segmented text prompts serve as high-level drivers of video progression.

**Long-context video generation.** While advances in autoregressive methods have allowed for generation of longer, and even potentially inifinite-length videos [14], the reliance on sliding context windows limit their effective memory. The straightforward approach of increasing memory by training with longer context lengths is effective but computationally demanding. Recent works have explored efficient architectural alternatives to attention-based transformers, such as state-space models and linear attention [69, 50]. While efficient, such methods often perform worse than conventional diffusion transformer architectures. Other concurrent works have proposed compressing prior frames to lower context length [86, 25], subsequently lowering computational demand at the cost of information lost in context frames. A separate branch of works ground previously generated frames in a 3D representation (e.g., point clouds [83, 26, 82, 57]) and query this representation based on the camera position of future frames. These methods allow for precise camera control, but struggle to handle dynamic scenes with complex motions. SlowFast-VGen [38] introduces a dual-branch slow–fast learning paradigm to explicitly disentangle long-term temporal dependencies from short-term motion dynamics, enabling more coherent and action-aware long video generation. Complementarily, [17] leverages continual adaptation at inference to maintain temporal consistency over extremely long sequences, demonstrating that lightweight test-time optimization can substantially extend generation duration without retraining.

In this work, we adopt the conventional approach of concatenating recent context frames as working memory in diffusion-based generation to autoregressively generate multiple future frames. Meanwhile, we iteratively predict and filter the static point maps of newly generated frames to update a global spatial memory, which serves as geometric guidance for long-term generation. We further incorporate a sparse set of historical reference frames as an episodic memory.

# 3 Method

Our framework includes mechanisms for storing new observations into memory and retrieving information from it. Drawing inspiration from three distinct forms of human memory—***spatial, working, and episodic***—we introduce a memory storage mechanism that models each type using a dedicated representation. In the following section, we describe how each memory type is constructed and maintained (Sec. 3.2), how it is integrated into the model as a conditional signal (Sec. 3.3), and how we curate the appropriate data to facilitate efficient learning of these mechanisms (Sec. 3.4).

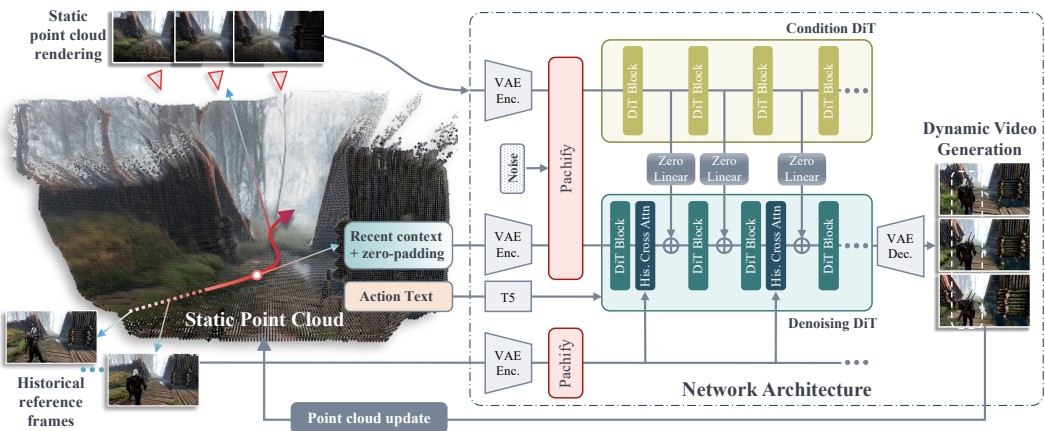

Figure 2: **Overview of our system.** A latent video generation model, implemented by a diffusion transformer (DiT), is conditioned on three different memory mechanisms when autoregressively generating new frames. First, recent context frames model a *short-term working memory*. Second, a point cloud representation (left) is autoregressively generated along with the video frames. This *long-term spatial memory* contains the static parts of the world. Third, a set of historical reference frames (lower left) is stored as a *sparse, long-term episodic memory*. Together, these memory mechanisms enable consistent long-term video generation.

## 3.1 Preliminaries

Diffusion models [59, 36, 61] learn to model a data distribution by reversing a forward diffusion process denoted by $x_t = \alpha_t x_0 + \sigma_t \epsilon$, where the positive scalar parameters $\alpha_t$ and $\sigma_t$ determine the signal-to-noise ratio based on a predefined noise schedule, and $\epsilon$ is drawn from a standard Gaussian distribution. The diffusion model then learns to predict the noise added through the following denoising objective:

$$\mathcal{L}(\theta) = \mathbb{E}_{t,x_0,\epsilon} \left\| \epsilon_\theta(x_t, t) - \epsilon \right\|_2^2. \tag{1}$$

Video diffusion models commonly operate in a latent space, following a two-stage process: first, input videos are encoded using a 3D variational autoencoder (VAE) [66, 27, 10], and then a diffusion model is trained to model the resulting latent representations. Generation proceeds by sampling within the latent space and decoding the results back to pixel space. Our prototype implementation builds on CogVideoX [79], which adopts this two-stage framework. Specifically, CogVideoX employs a diffusion transformer with 3D attention blocks to capture the distribution of the latent space. Our model improves upon this architecture with additional control signals for enabling long-term memory.

## 3.2 Spatial Memory Storage

**Coherent Static Structure for Spatial Guidance.** Human *spatial memory* refers to our ability to encode, store, and retrieve information relating to the physical layout and structure of our environment. To this end, we construct a persistent spatial memory in the form of a long-term static point map that captures high-confidence, temporally consistent 3D structures. To further separate static elements like buildings from dynamic elements like characters and animals, we adopt truncated signed distance function (TSDF) fusion [84]. We denote the TSDF value and associated weight of a voxel $v$ as $D(v)$ and $W(v)$, respectively. Given a new observation from frame $i$, the voxel update rule follows the standard weighted averaging:

$$D'(v) = \frac{W(v) \cdot D(v) + w_i \cdot d_i(v)}{W(v) + w_i}, \quad W'(v) = W(v) + w_i, \tag{2}$$

where $d_i(v)$ is the truncated signed distance between voxel $v$ and the observed surface in frame $i$, and $w_i$ is a frame-dependent confidence weight (typically set to 1).

This fusion process inherently filters out dynamic elements in the scene: due to inconsistent depth observations across frames, such voxels accumulate low-confidence, noisy TSDF values and are naturally suppressed in the final fused volume.

During the autoregressive generation process, spatial memory is incrementally updated with newly observed static maps, which are reconstructed in an online recurrent manner by CUT3R [70] and filtered by TSDF-Fusion to eliminate the dynamic parts.

**Recent Frames for Dynamic Context Guidance.**   In humans, ***working memory*** is in charge of temporarily holding information needed for performing reasoning and comprehension tasks. Similarly, our model requires knowledge of nearby previous frames to generate temporally coherent future frames. Drawing on this concept, we incorporate a short-term memory stream based on recent frames, which provides motion continuity atop the static scene structure. We adopt a simple autoregressive generation strategy, where the model generates $N - k$ future frames by conditioning each step on the most recent $k + 1$ latent frames. This procedure can be iteratively applied, enabling open-ended video generation with consistent temporal dynamics.

**Representative Historical Slots to Enhance Details.**   Human *episodic memory* is a type of explicit memory that stores specific important events from the past, allowing us to "recall" the relevant experiences when needed. While the fused static point cloud captures stable scene geometry, the fused static point cloud is often too sparse to preserve detailed visual cues from the past. To compensate for this, we maintain a set of representative historical frames as auxiliary references. Specifically, during generation, we monitor the size of newly revealed unknown regions via mask-based visibility checks. When the revealed area exceeds a predefined threshold, the corresponding frame is selected and added to the memory set in an incremental fashion.

## 3.3  Memory-guided Video Generation

Different from those video generation models that focus on camera control under the same temporal sequences [82, 7, 6, 83, 26], we mainly focus on the temporal progression. For example, for a car driving on the road, when the camera moves to the left, the car should complete reasonable dynamic motion in accordance with the hints of the camera language and the input prompt, while maintaining the consistency of the static part to achieve the simulation of the interactive world model. Based on this requirement, we carefully design several key modules to achieve dynamic and static decoupling.

As illustrated in Figure 2, first, we introduce static point cloud rendering as an additional conditioning input to our video diffusion model. The condition video is rendered from the current static spatial memory along the input trajectory, with background regions lacking point clouds set to black. We then utilize the pre-trained 3DVAE [79] to encode the static point cloud rendering into condition latents. We follow a similar design as the ControlNET [87] to add the static point clouds rendering to guide the camera movement and keep the static areas consistency. We copy the first 18 pre-trained DiT blocks from CogVideoX as the condition DiT to process the condition latents. In the condition DiT, we process the output feature from each main DiT block through a zero-initialized linear layer before adding it to the corresponding feature map in the main DiT. Second, to support the generation of new dynamic elements and the temporal extension of existing ones, we propose to concatenate the last five frames of source video tokens with the target video tokens along the frame dimension for dynamic context guidance. In addition, the target condition tokens are also combined with recent context tokens as mentioned above to ensure frame-level correspondence. Third, for modeling information exchange between memory frames and the frames currently being generated, we select the representative historical slots frames as auxiliary reference frames. This reference frames are also encoded by 3DVAE and patchify them as reference tokens. we add a historical cross attention to guide information exchange between the frames currently being generated and memory frames. Specifically, the video tokens act as queries and the reference tokens serve as keys and values.

## 3.4  Geometry-grounded Video Dataset Creation

To train and evaluate our geometry-grounded video generation model, we require a custom dataset as shown in Figure 3 which is described in the following.

**Dataset Construction.**   We build our dataset from raw videos collected from MiraData [42], segmenting each video into multiple 97-frame clips. For each clip, the first 49 frames serve as the source sequence and the remaining 48 as the target, with a shared transition frame between the source and target sequence to preserve temporal continuity. To recover scene geometry, we perform

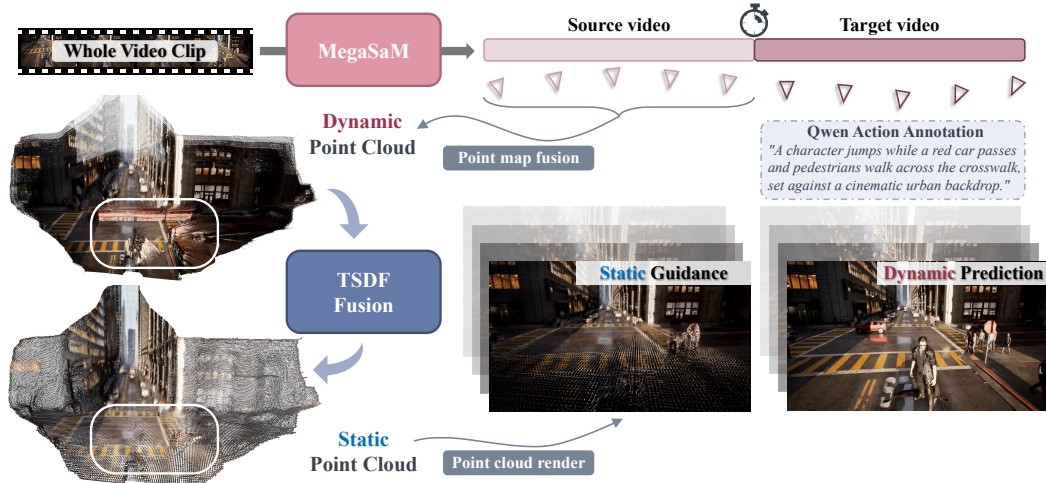

Figure 3: **Dataset construction pipeline.** We use Mega-SaM [47] to extract camera poses and dynamic point maps from the full video clip. For the source part, *dynamic regions are erased* via TSDF-Fusion, and the point cloud is rendered along the target trajectory to to serve as static geometry guidance for the target part. Qwen [77] generates annotations for actions in future target frames.

Table 1: **Quantitative evaluation.** We evaluate our method and baselines using FVD, view recall, and a user study. For view recall, we use standard metrics to compare the consistency of revisiting parts of a scene. The user study provides relative average human ranking scores for three different criteria of all baselines. Our method outperforms these baselines in all cases.

| Method | FVD | View Recall Consistency | | | User Study | | |
|---|---|---|---|---|---|---|---|
| | | PSNR ↑ | SSIM ↑ | LPIPS ↓ | Cam-Acc ↑ | Stat-Cons ↑ | Dyn-Plaus ↑ |
| TrajectoryCrafter | 355.23 | 11.71 | 0.4380 | 0.5996 | 1.6320 | 1.7802 | 1.6255 |
| DaS | 363.36 | 12.01 | 0.4512 | 0.5874 | 2.5660 | 2.4396 | 2.7033 |
| Wan2.1-Inpainting | 280.06 | 12.16 | 0.4506 | 0.5875 | 2.1760 | 2.3956 | 2.2701 |
| Ours | **157.11** | **19.10** | **0.6471** | **0.3069** | **3.6260** | **3.3846** | **3.4011** |

4D reconstruction using Mega-SaM [47], extracting camera intrinsics, extrinsics, and per-frame depth maps. We apply TSDF-Fusion to the source frames, integrating RGB-D observations into a volumetric grid. This process suppresses inconsistent depth caused by dynamic objects, yielding a clean reconstruction of the static scene.

**Paired Training Data.** Given the fused geometry, we project the target camera poses to render visibility masks and static-region reconstructions via point-based rendering. The full RGB frames of the target sequence are retained as future supervision, containing dynamic elements beyond the static scene memory. Our final dataset comprises 90K structured video samples, each paired with explicit 3D spatial memory and future observations.

Our dataset can be downloaded here [1]. Additional details on memory storage and retrieval mechanisms as well as dataset creation are found in the supplement.

## 4 Experiments

**Implementation Details.** We implement our conditional video diffusion model based on CogVideoX-5B-I2V [79] architecture, pretrained from DaS [26]. During training, we set the video length to 49 frames with a resolution of $480 \times 720$. We trained for 6,000 iterations with a learning rate of $2 \times 10^{-5}$, using a mini-batch size of 8 and are conducted on eight NVIDIA-A100 GPUs. At inference time, we adopt the latest 5 historical frames from the recent sequence to enable smooth

---
[1] https://huggingface.co/datasets/ysmikey/spmem_megadata

Table 2: **Quantitative evaluation on VBench**. Our model achieves top overall performance among relevant baselines, as measured by VBench metrics.

| Method | Aesthetic Quality ↑ | Imaging Quality ↑ | Temporal Flickering ↑ | Motion Smoothness ↑ | Subject Consistency ↑ | Background Consistency ↑ |
|---|---|---|---|---|---|---|
| TrajectoryCrafter | 0.5255 | 0.6428 | 0.6160 | 0.9843 | 0.8830 | 0.9227 |
| DaS | 0.5635 | 0.6617 | 0.7520 | 0.9856 | 0.9325 | 0.9494 |
| GEN3C | 0.5203 | 0.5654 | 0.7179 | 0.9882 | 0.9178 | 0.9433 |
| Wan2.1-Inpainting | 0.5661 | **0.6788** | 0.6433 | 0.9868 | 0.9357 | **0.9513** |
| Ours | **0.5835** | 0.6701 | **0.7580** | **0.9886** | **0.9359** | 0.9506 |

motion prediction. At each auto-regressive iteration, the point map of the newly generated frames are predicted, aligned, and fused into the historical global point clouds, where we create new point rendering sequence given the aimed camera trajectory.

**Metrics and Baselines.**    Our evaluations primarily focus on baseline methods that use point-map-based conditioning. This includes TrajctoryCrafter [82], DiffusionAsShader (DaS) [26], GEN3C [57], and also the state-of-the-art video generative model Wan2.1 [68]. Our test set includes 500 randomly selected video sequences from MiraData, which are not seen during training. We evaluate each method on FVD, view recall consistency, and general video quality on multiple dimensions. Specifically, 1) For static spatial information, the view recall consistency is evaluated using image reconstruction metrics (i.e., PSNR, SSIM, and LPIPS) for paired frames at the same camera location within a video sequence generated with forward and reversed camera trajectory. 2) For general video quality evaluation, we use the standard FVD and six metrics from Vbench [39] for the visual quality, motion smoothness, and consistency. Moreover, we conduct a user study to further validate the criteria above.

## 4.1    Quantitative Evaluation

**Evaluating View Recall Consistency and Camera Accuracy.**    To verify that our method benefits from the spatial memory mechanism by maintaining high consistency and accurate camera control when revisiting previously generated parts of the world, we conduct experiments on reversed trajectories, where the same camera pose is visited twice, enabling the construction of paired data for reconstruction metrics. As shown in Table 1, our method achieves significantly improved scores in terms of FVD, PSNR, SSIM, and LPIPS compared to all baselines, owing to our memory mechanisms. It is worth noting, however, that even the PSNR of our method is far from perfect, indicating that remembering each and every visual detail of a complex scene is a very challenging task.

**Evaluation on Video Quality.**    We further utilize VBench to evaluate general video quality across multiple dimensions, as shown in Table 2. Compared with baseline methods, our approach demonstrates better performance in aesthetic quality, reduced temporal flickering, smoother motion, and improved subject consistency. Methods primarily designed for 3D static NVS or 4D NVS within the same temporal sequence perform less effectively on our benchmark, likely due to the sparsity of the retained static point cloud and the presence of large spatial holes. While Wan2.1 surpasses our method in imaging quality due to its advanced backbone, and also achieves a higher score in background consistency, we note that the Wan2.1 inpainting model often fails to follow geometric guidance and tends to generate relatively static scenes, which makes it easier to maintain high background consistency scores.

## 4.2    Qualitative Evaluation

We conduct qualitative comparisons with other geometry-grounded methods, focusing on three key criteria, as illustrated in Figure 4. First, our method demonstrates superior performance in accurately following camera trajectories, guided by point map rendering. In contrast, the Wan2.1 inpainting model and DaS model often fail, especially under significant camera motion. Second, for view recall consistency, we present pairwise comparisons between two frames generated at the same camera pose but at different points in the sequence. Compared with the baselines, our results exhibit significantly higher consistency in static regions during scene revisits, thanks to the static memory mechanism. Finally, we evaluate the ability to generate new actions based on instructions while incorporating static memory. We focus on the harmonious integration of static and dynamic elements, as well as

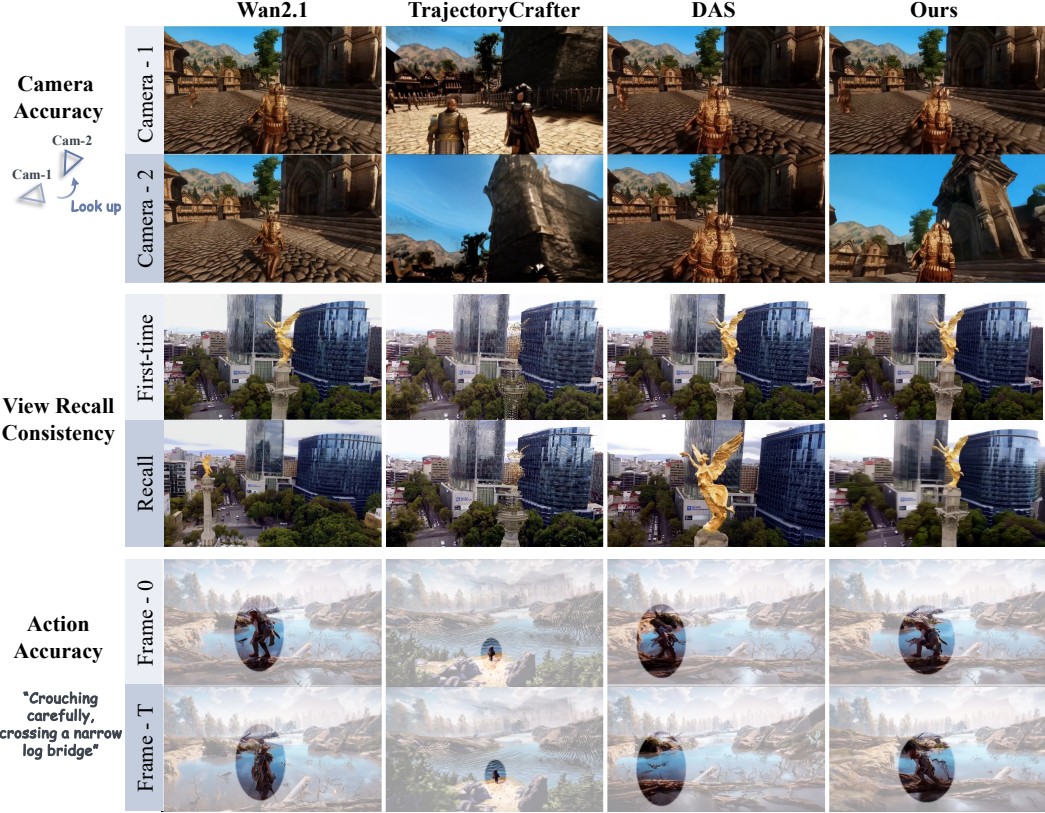

Figure 4: **Qualitative evaluation.** We compare our approach to relevant baselines in several conditions. Baselines cannot accurately generate significant camera pose changes while maintaining a consistent scene (top). When revisiting a previously seen camera pose, baselines fail to complete sparse point clouds or forget details, resulting in inconsistency (center). The accuracy of prompted actions is often low, and sometimes the character disappears during the generation for the baselines (bottom). Our approach successfully handles these challenging scenarios.

how closely the dynamic components follow the instructions. The comparison shows that our method performs well in action prediction, whereas the others either fail to accurately follow the instructions or suffer from action drifting, severe deformation, or even character disappearance.

## 4.3 User Study

We selected 14 representative use cases, including novel-view synthesis of static scenes, novel-view synthesis of dynamic scenes with temporal progression of dynamic subjects in first-person and third-person perspectives, and scene styles covering realistic and game style. We conduct a user study, evaluating baselines and our methods from three perspectives: camera accuracy(Cam-Acc), static consistency(Stat-Cons), and dynamic plausibility (Dyn-Plaus). We invited 20 subjects, each with at least one year of experience in video/3D/4D generation, to rank the results generated by the four methods (TrajectoryCrafter, DaS, Wan2.1-Inpainting, and ours). Following ControlNet, we evaluate the results using the Average Human Ranking (AHR) metric, where participants rated each output on a scale of 1 to 4 (with lower scores indicating poorer quality). The average rankings in Table 1 (right) show that our method achieves clear and consistent improvements over the baselines across all metrics.

## 4.4 Ablation Study

To verify the effectiveness of each memory component in our video generation framework, we conduct comprehensive ablation studies, as shown in Table 3. Experimental results on VBench metrics indicate that each component consistently contributes to performance improvements. Unsurprisingly,

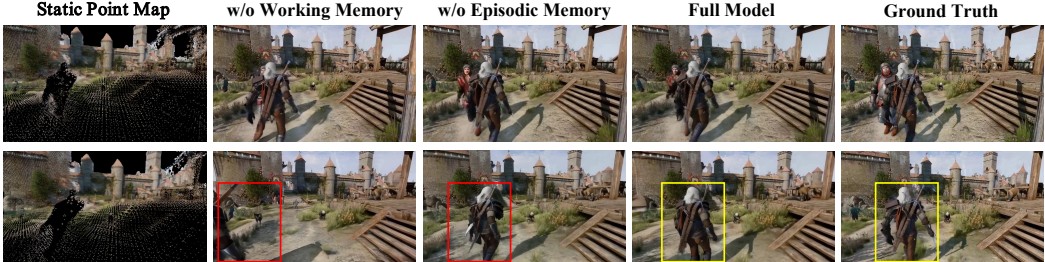

| Static Point Map | w/o Working Memory | w/o Episodic Memory | Full Model | Ground Truth |

Figure 5: **Ablation of different memory mechanisms.** We evaluate several variants of our model: **w/o short-term working memory**: the full model without recent context frames; **w/o long-term episodic memory**: the full model without sparse historical keyframes; **full model** including short-term working and long-term spatial and episodic memory. Unsurprisingly, the working memory is required for smooth and plausible motions of dynamic objects. The episodic memory is crucial in helping remember visual details from the past, including previously seen characters or objects.

Table 3: **Ablation of memory mechanisms.** Using all three types, i.e., short-term working and long-term spatial and episodic memory, leads to the best results measured by VBench metrics.

| Method | Aesthetic Quality ↑ | Imaging Quality ↑ | Temporal Flickering ↑ | Motion Smoothness ↑ | Subject Consistency ↑ | Background Consistency ↑ |
|---|---|---|---|---|---|---|
| w/o episodic mem | 0.5603 | 0.6485 | 0.7260 | 0.9870 | 0.9326 | 0.9489 |
| w/o working mem | 0.5551 | 0.6384 | 0.6740 | 0.9862 | 0.9331 | 0.9453 |
| Full model | **0.5835** | **0.6701** | **0.7580** | **0.9886** | **0.9359** | **0.9506** |

the context frames play a crucial role in enhancing short-term motion coherence. During the autoregressive generation process, the context frames enable smooth transitions in dynamic regions and help produce more plausible motions consistent with the preceding frames. The sparse set of historical reference frames enable the model to better retain and utilize temporally distant details. This episodic memory improves long-term consistency for static regions and subjects, and further enhances the plausibility and continuity of motions involving moving entities. The best results are achieved with both of these mechanisms as well as our long-term spatial memory. This is evidenced by both Table 3 and Figure 5, showing that each of our memory mechanisms significantly improves the model's quality in terms of motion smoothness, static consistency, and overall visual quality.

## 5 Discussion

Inspired by the mechanisms of human memory, we introduce a geometry-grounded long-term spatial memory mechanism for video world models. This mechanism improves quality, spatial consistency, and context length compared to relevant baselines.

**Limitations and Future Work.** The TSDF-Fusion algorithm we use for storing newly generated information into the spatial memory is far from perfect. Specifically, artifacts are introduced when looking at previously generated content from camera poses that are very different from those of the previous observations, as illustrated in Figure 6. Our memory mechanism is primarily designed to enable *spatial consistency* whereas frame packing strategies for extending the temporal context window size [86] primarily focus on *character consistency*. Future work may combine these mechanisms to achieve both types of consistency. The forgetting problem we tackle is just one of several challenges of video world models. Drift, or image quality degradation due to error accumulation over time, is another challenge that we do not address.

**Societal Impacts.** Video generation models provide significant benefits for content creation but could be adapted for DeepFake generation. Such applications pose significant societal risks and we strongly oppose the use of our work to create deceptive content intended to mislead or spread misinformation.

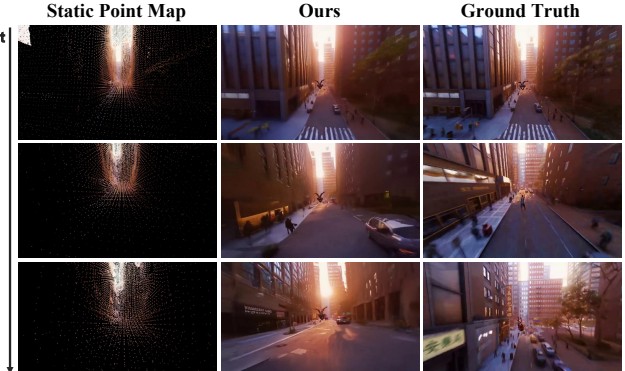

Figure 6: **Failure case.** When the distance between consecutive camera poses is too large and the trajectory exhibits overly abrupt angles, the 4D reconstruction may fail, resulting in significant ghosting artifacts between frames. Consequently, TSDF-Fusion will filter out a large portion of point clouds that should belong to static regions, ultimately leading to an extremely sparse spatial memory and loss of critical information. For example, Spiderman rapidly traversing between skyscrapers illustrates how such a challenging camera trajectory can cause omissions in spatial memory storage, resulting in imprecise camera control and inconsistencies.

**Conclusion.** Video world models play a crucial role for content creation or creating training data for agents or robots. Enabling long-term consistency through memory mechanisms like ours makes these models more effective.

# 6 Acknowledgment

We thank Google and Shanghai Artificial Intelligence Laboratory for their support. Ryan Po is supported by a Stanford Graduate Fellowship.

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
