# OpenReview forum: "Video World Models with Long-term Spatial Memory"
_NeurIPS.cc/2025/Conference — NeurIPS 2025 poster_

### Official Review · Reviewer_e9qv · 2025-06-30

**Clarity:** 2
**Significance:** 2
**Originality:** 2
**Rating:** 5
**Confidence:** 3

**Summary:**

Paper presents a world model system (predicts video frames) that is incorporated with spatial and keypoint memory to address challenges with persistent memory with world model image generation. The presented method outperforms baselines both in quantitative metrics (video quality, user studies, consistency) as well as in qualitative sense.

**Questions:**

1) What are the upper/lower bounds of the metrics in the Table 1, Table 2 and Table 3, and what is their rough meaning? For Table 1, you could consider including baseline results (e.g., only repeating static image) and oracle/upper bound (e.g., the original video clip but passed through the image encoder/decoder). This way you can could tell how much of the sequence error comes from the image encoding/decoding.

2) Did you repeat the experiments multiple times and then averaged over to report statistics of results? It is unclear how sensitive the model is to training, and if the results are meaningful, primarly in Table 2 and Table 3, where the differences seem very minute.

3) How was the testing data sampled? L211 states it is "500 randomly selected video sequences from MiraData", but it is unclear if these samples are from unseen video clips. Were some raw videos (as used in L186) left out for test set, or are test set clips from same set as training clips? If test clip is from a middle of recording, having the immediate past and future in training set might make the task easier. It would be interesting to see the results on unseen videos, or test samples taken from the very and or beginning of the raw videos.

4) How were human rankings collected, exactly? Could you share the forms you used, and more information of the participants of the study (e.g., how were they recruited). The way questions are posed can have a big influence on the results (e.g., do you show oracle/groundtruth results [should get 4], do you only show one video at a time, etc).

5) Would it be possible to see view recall consistency quantitative results (Table 1) for the ablation? I think these are important metrics for the ablations, especially the one studying impact of the episodic memory.

**Ethical Concerns:**

["NO or VERY MINOR ethics concerns only"]

**Final Justification:**

Authors have addressed my concerns and I have increased my original rating 4 to 5 (Accept) to signal my vote to accept the paper.

Paper solution is strong over the baselines across several sets of data, and the experiments were conducted well. The main limitations of paper are the core methodology being slightly complex with multiple parts, and the lack of deeper understanding what helps and where.

**Limitations:**

See strengths and weaknesses.

**Paper Formatting Concerns:**

-

**Quality:**

2

**Strengths And Weaknesses:**

I vote for a low confidence accept (hence borderline). The paper presents positive results, both quantitative and qualitative, and the method is novel/interesting enough to be of interest for NeurIPS community, and I can see other works building on this. However, the main weaknesses of the paper are lack of theoretical insight/analysis (now answers to "why" questions), and the fact that the results are intuitively expected (enforcing memory in video generation process improves video generation consistency).

I have not kept up with the latest improvements in the field, so I may have missed something. In addition, I have some questions (see Questions) that should be addressed for a more confident vote.


**Quality**

- +Reported performance is higher than baselines, in both quantitative (image quality and consistency) and qualitative metrics (human judges)
- +The work contributes a custom dataset
- /Comparison against three baselines, but Wan2.1 is not directly comparable (does not have action conditioning)
- -No theoretical/insightful analysis. While I appreciate the ablation studies on the importance of the provided parts, they do not provide answers to "why" they improve results, apart from the intuition.
- -Method was tested with custom dataset/benchmark (data created from MiraData), which is required for the data method uses, but the data may be out of the distribution of baselines (or the data they were originally presented in). The results would be more convincing if the benchmark was established/defined in a prior work.


**Clarity**

- -The text contains numerous cases of vague/unfounded adjectives. These do not bring more value to the scientific text as is, or may even confuse, so they should be revised. For example:
  - L253. "Comprehensive" (what makes it comprehensive?)
  - L260. "Large margin". What is a large margin? The more scientific method is to perform statistical test on the results.
  - Figure 6 caption: "_swiftly_ swinging"
  - Section 4.2: while qualitative, this section too has many vague adjectives ("fail to _closely_", "significantly higher consistency", "harmonious").
- L188. What is "shared transition frame"?



**Significance**

- +Dataset and the approach can be useful building blocks/comparisons for future work. I am particularly interested in the evolution of different approaches (e.g., raw image-to-image generation vs. point-cloud generation like here).
- -While promising results, they are not very surprising: the problem is long term video generation, which requires remembering objects in the scene. The paper proposes very explicit way of handling this (storing keyframes, explicit mapping to point cloud and back), which improves results. I do not expect the "insight" of the work to be very valuable to the NeurIPS community.

**Originality**

I am not deeply familiar with the literature, but to me, I not seen major works with similar approach. However, the method seems to be a continuation of TrajectoryCrafter, both relying on point clouds to have consistency, but this proposed method takes this further by encoding the memory.

---

> ### Author Rebuttal · Authors · 2025-07-31
>
> Thank you for your detailed review and insightful comments. We will address your questions and concerns below and in the revised paper.
>
> ---
>
> ### W1: Wan2.1 is not directly comparable.
>
> Thanks for the feedback. We make slight adaptations to the Wan2.1-I2V so that it could be comparable in the context. Specifically, we use static point cloud rendering to initialize video latents to implement camera control like trajectorycrafter, and add a low level of noise on top. To ensure the motion coherence of dynamic elements, we select the recent frames as the reference frames. Based on the original backbone of Wan2.1-I2V,  we perform dual encoding on reference frames through Clip visual encoder and 3D-VAE respectively: the semantic features encoded by Clip-vision are concatenated with text prompts to maintain content consistency, and the spatial features extracted by 3D-VAE are concatenated with video latents to enhance geometric continuity, finally achieving a temporal progressive NVS task similar to ours.
>
> ---
>
> ### W2: Lack of theoretical/insightful analysis.
>
> Thank you for the thoughtful comment. The working memory primarily derives from recent context frames. These clean reference contexts are concatenated with noisy video latents at the frame level, enabling us to adapt the self-attention layers into reference mixing layers that effectively capture cross-view dependencies. This architecture not only enhances 3D consistency but also improves short-term motion coherence; The episodic memory system mainly derives from historical reference key frames. Compared to working memory, these views originate from more distant camera positions, endowing the model with stronger robustness. Particularly when user-input trajectories exhibit "revisit" (overlaps with previously generated regions), our designed historical cross-attention mechanism can effectively retrieve and leverage these stored details. This approach ensures long-term consistency while preventing memory forgetting, as the system maintains access to relevant historical information even during novel view synthesis. We will add the analysis to the final version.
>
> ---
>
> ### W3: More convincing testset setup.
>
> Thank you for the insightful feedback. We are unable to directly adopt the benchmark established in prior work, as the task definitions are not fully aligned and their dataset is incompatible with the requirements of our setting. But in order to mitigate the impact of the domain gap, we construct a new subset of 100 highly dynamic scenes covering both real-world and game-engine domains, including rainy and snowy weather as well as day and night environments from Sekai [1], a large-scale video world exploration dataset. Specifically, for the reconstruction metrics, we far surpass other methods, which reflect our ability to keep 3D consistency and control camera accuracy. In terms of vbench metrics, we show superior video quality, although aesthetic quality and imaging quality lag slightly behind Wan2.1-inpainting, as mentioned in Sec.4.1 of our paper, this is due to their advanced backbone. Overall, our method still shows robust performance in these difficult and OOD cases.
>
> [1] Li, Zhen, et al. "Sekai: A Video Dataset towards World Exploration." *arXiv preprint arXiv:2506.15675* (2025).
>
> | Method            | PSNR↑       | SSIM↑      | LPIPS↓     | FVD↓       | Aesthetic Quality | Imaging Quality | Temporal Flickering | Motion Smoothness | Subject Consistency | Background Consistency |
> | ----------------- | ----------- | ---------- | ---------- | ---------- | ----------------- | --------------- | ------------------- | ----------------- | ------------------- | ---------------------- |
> | TrajectoryCrafter | 9.8791      | 0.3497     | 0.5892     | 862.73     | 0.4846            | 0.5907          | _0.7158_              | 0.9866            | 0.9533              | 0.9425                 |
> | DaS               | _10.0074_     | _0.3849_     | _0.5719_     | 962.46     | 0.5102            | 0.6272          | 0.6421              | 0.9849            | 0.9565              | 0.9508                 |
> | Wan2.1-Inpainting | 10.0013     | 0.3811     | 0.5797     | _834.30_     | **0.5151**        | **0.6709**      | 0.6276              | _0.9893_            | **0.9583**          | **0.9668**             |
> | Ours              | **19.0984** | **0.6789** | **0.2228** | **256.18** | _0.5123_            | _0.6347_          | **0.8526**          | **0.9903**        | _0.9570_              | 0.9602                 |
>
> (Bold is best, italic is the second.)
>
> ---
>
>
> ### W4: Issues in clarity.
>
> Thank you for pointing this out. We will carefully review the writing and revise the final version to avoid vague or unsupported adjectives.
>
> ---
>
>
> ### W5: Insight and originality of the work.
>
> Beyond the spatial memory mechanism, another major difference from TrajectoryCrafter and other point cloud-conditioned methods is our ability to handle temporally misaligned inputs and outputs. Specifically, while TrajectoryCrafter performs dynamic novel view synthesis with temporally aligned inputs and outputs, our method predicts future observations based on static conditions and dynamic references stored in working memory. This goes beyond a simple inpainting task. We believe that the problem setup, the construction of corresponding data pairs, and the proposed model design will be inspiring to the community.
>
> ---
>
>
> ### Q1: Metric clarification.
>
> In Table 1, user study scores range from 1 to 4, while reconstruction metrics (PSNR, SSIM, LPIPS) mainly assess camera accuracy and 3D consistency in static regions. Because our model uses a 3D VAE for video encoding/decoding, these metrics are particularly sensitive—compression-like processing can introduce measurable deviations even when static regions appear nearly identical to human observers. Following your suggestion, we tested repeating static videos and found approximate ranges: PSNR ↑ (0–22), SSIM ↑ (−1 to 0.75), LPIPS ↓ ( 0.15 to +∞).
>
> While the metrics in Tables 2 and 3 are from VBench, a widely used video generation benchmark. According to the VBench paper, all metrics are normalized to 0–100%, where 100% is ideal and lower bounds indicate worst case. Aesthetic Quality (0–1.0) measures artistic merit, Imaging Quality (0–1.0) detects blur/noise, Temporal Flickering (0.6293–1.0) gauges frame consistency, Motion Smoothness (0.7060–0.9975) checks physics-compliant motion, Subject Consistency (0.1462–1.0) tracks foreground stability, and Background Consistency(0.2615–1.0) evaluates scene coherence.
>
> ---
>
>
> ### Q2: Additional experimental setup.
>
> Following the widely adopted experimental setup in previous video generation benchmarks, we fix a single random seed for all experiments and report the corresponding results. We also observe that running the experiments with different seeds and averaging the results over the full 500-sample test set leads to only negligible variations.
>
> ---
>
>
> ### Q3: Testset setup.
>
> The 500 test cases are sampled entirely from unseen videos. Specifically, the raw videos have been pre-split into disjoint training and testing sets, ensuring that test clips and training clips never come from the same original video.
>
> ---
>
>
> ### Q4: User study setup.
>
> In the final version, we will include screenshots of the user study questionnaire and provide detailed user statistics for clarification. Briefly, we invited 13 graduate students working on video generation research to participate in the user study. Each participant was presented with ground truth and generated videos for evaluation. We report the average human ranking scores based on their responses.
>
> ---
>
>
> ### Q5: Quantitative results of view recall consistency for the ablation.
>
> Thanks for the insightful suggestion. We report the quantitative results of view recall consistency for the ablation as below. Our full model, which uses short-term working and long-term spatial and episodic memory, performs best on reconstruction metrics, demonstrating the effectiveness of the memory mechanism in keeping long-term 3D consistency.
>
> | Method           | PSNR ↑      | SSIM ↑     | LPIPS ↓    |
> | ---------------- | ----------- | ---------- | ---------- |
> | w/o episodic mem | 18.8665     | 0.6628     | 0.2898     |
> | w/o working mem  | 10.1379     | 0.4008     | 0.5682     |
> | Full model       | **19.0984** | **0.6789** | **0.2228** |

---

> > ### Comment · Reviewer_e9qv · 2025-08-04
> >
> > I thank the authors for exhaustive coverage of the limitations and questions. Most of my concerns have been addressed (test set quality, human data collection, additional experiments, metric ranges), and I believe the paper is a good contribution to NeurIPS, and I am increasing my score by one from 4 to 5 (Accept). Further increase of score would require more fundamental changes (e.g., simpler setup, more insight on what is happening).

---

> > > ### Author Response · Authors · 2025-08-05
> > >
> > > We sincerely thank you for your thoughtful comments and constructive questions. We are especially grateful for your engagement during the rebuttal stage and truly appreciate your willingness to raise your rating. We will make sure to incorporate the information in the rebuttal into the final version of the paper.

---

### Official Review · Reviewer_Kmfd · 2025-06-30

**Clarity:** 4
**Significance:** 3
**Originality:** 3
**Rating:** 5
**Confidence:** 5

**Summary:**

This paper proposes a video-gen method that can condition on reconstructed point map and historical frames, thereby maintaining episodic, working and spatial memory for long video generation.

**Questions:**

Please see the above.

**Ethical Concerns:**

["NO or VERY MINOR ethics concerns only"]

**Limitations:**

Please see the above.

**Paper Formatting Concerns:**

no formatting issues

**Quality:**

3

**Strengths And Weaknesses:**

Thanks for the good paper.
* Memory is always an important perspective for video-gen
* Good formulation that involves spatial, working and episodic memory.
* Results and ablative experiments are good.

Several concerns and questions:

* **Related Works** Some related papers [1,2] use test-time training to maintain memory. Please cite. [1] also mentioned episodic memory in the paper.

[1] SlowFast-VGen: Slow-Fast Learning for Action-Driven Long Video Generation

[2] One-Minute Video Generation with Test-Time Training

* **Explicit VS Implicit 3D representations** The paper mentioned the spatial memory of human beings. However, the place cells and grid cells are more implicit representations than registered pointmap. Do you think in the future we can instead utilize more implicit 3D representations (e.g., similar to [1] and [2] where the latents might potentially already encode part of the 3D representations?) Explicit 3D reconstruction can be accurate but also quite heavy sometimes, especially when the scene gets very large. What do you think are the pros & cons of both? No experiments needed just hope to hear the authors' opinions.
* **3D reconstruction method** The authors use TSDF for 3D reconstruction. Have you experimented with VGGT or other 3D reconstruction methods? Can you share why did you choose TSDF among other, to give us some insights? No experiments needed just hope to hear the authors' opinions.
* **Dynamic humans / other agents in static scenes** In practice, when we utilize 3D reconstruction methods such as VGGT on dynamic scenes, the moving humans can be a big problem, which can result in scene distortion or occlusion. Have you come up with any specific methods to deal with this, or do you think it's not a problem?
* **More related works** These papers are relevant and worth reading, though not citing them is okay as they are not in the same field. Specifically, WonderPlay deals with the dynamic actions mentioned in the last limitation. [4] also talks about 3D working and episodic memory.

[3] The wonder series (WonderJourney, WonderWorld, WonderPlay)

[4] 3DLLM-Mem: Long-Term Spatial-Temporal Memory for Embodied 3D Large Language Model

---

> ### Author Rebuttal · Authors · 2025-07-31
>
> Thank you for your detailed review and insightful comments. We will address your questions and concerns below and in the revised paper.
>
> ---
>
> ### Q1&Q5: Include more related works.
>
> Thank you for the thoughtful suggestions. We will cite the related works [1-4] and include proper discussions in the final version.
>
> ---
>
> ### Q2: Explicit VS Implicit 3D representations.
>
> Thank you for the insightful and valuable feedback. Both explicit and implicit 3D representations have their respective advantages and limitations. Explicit representations (e.g, point clouds) offer high precision, easy editability, and strong compatibility, making them ideal for scenarios requiring strict geometric control, though they come with higher storage and computational costs than implicit representation. In contrast, implicit representations excel in efficient compression and scalability, but face challenges in interpretability and training complexity. We believe the future lies in hybrid representations, for instance, using implicit encoding for global semantics while employing explicit structures for local detail processing.  We will continue to explore this direction in our future research.
>
> ---
>
> ### Q3: 3D reconstruction method.
>
> We leverage Mega-SaM during dataset construction and CUT3R during inference for 3D reconstruction. Since their outputs are frame-wise point maps, we further apply TSDF-Fusion to aggregate them into a global point cloud and filter out dynamic elements such as characters and animals. This combination of state-of-the-art 3D reconstruction methods with the traditional TSDF-Fusion approach enables efficient and accurate camera pose estimation and global point cloud reconstruction.
>
> ---
>
> ### Q4: Dynamic humans / other agents in static scenes.
>
> Thank you for the thoughtful comment. Dynamic scenes indeed pose significant challenges. As mentioned in Q3, we chose Mega-SaM and CUT3R in place of VGGT as our main reconstruction tools, as both are specifically designed and trained for dynamic scene reconstruction to ensure accurate camera poses and point maps. However, directly combining per-frame point maps leads to artifacts and misalignment of dynamic elements. To address this, we further incorporate TSDF-Fusion to filter out dynamic content and obtain a clean, static global point cloud.

---

### Official Review · Reviewer_ddFV · 2025-07-02

**Clarity:** 3
**Significance:** 3
**Originality:** 3
**Rating:** 5
**Confidence:** 4

**Summary:**

The paper proposes Video World Models with Long-term Spatial Memory, combining

1. Short-term “working-memory” (recent frames),

2.  A geometry-grounded static point-cloud map updated online with TSDF fusion, and

3.  A sparse episodic key-frame buffer. The authors build a 90 K-clip “geometry-grounded” dataset derived from MiraData and show gains over TrajectoryCrafter, DaS, and Wan 2.1-Inpainting on view-recall metrics, VBench scores, and a 20-person user study.

**Questions:**

1. Please address the weaknesses mentioned.

2. Will the full 90 K-clip dataset (videos, depth, poses) be publicly released?

3. Did you retrain the baselines on your dataset?

4. Can you please report FVD for your method and compare against baselines?

5. What is the wall-clock time per generated 1-second video (e.g., 30 frames) and GPU memory footprint?

6. How does performance degrade when the depth estimator (CUT3R) produces noisy maps (e.g., night scenes, rain)?

**Ethical Concerns:**

["NO or VERY MINOR ethics concerns only"]

**Final Justification:**

The authors have addressed all my concerns, and I am satisfied. Hence I have increased my rating.

**Limitations:**

The authors have adequately discussed limitations.

**Paper Formatting Concerns:**

No formatting issues.

**Quality:**

3

**Strengths And Weaknesses:**

Strengths:

• The paper poses a timely problem: long-horizon drift/forgetting is a real blocker for autoregressive video world models.

• Straight-forward implementation– integrating point-cloud renders via a ControlNet-style branch is practical.

• The architectural diagram is clear. Clear writing style.

Weaknesses:

• Limited comparisons with recent works like Gen3C (Ren et al. CVPR 2025).

•  Metric choice biases towards static consistency. The main quantitative gain is PSNR/SSIM on revisited static frames; dynamic realism and action accuracy are mostly scored by the same subjective study. No perceptual metrics such as FVD.

---

> ### Author Rebuttal · Authors · 2025-07-31
>
> Thank you for your detailed review and insightful comments. We will address your questions and concerns below and in the revised paper.
>
> ---
>
> ### W1: Comparisons with recent works.
>
> Thank you for the valuable suggestion. We will include Gen3C and more recent works as additional baselines in our final version.
>
> ---
>
> ### W2&Q4: Metric choice biases and to report FVD.
>
> Thank you for the constructive feedback. For evaluating dynamics and actions, we follow prior works by reporting metrics such as motion smoothness from VBench, which is one of the most widely used indicators of motion quality. We would be happy to include additional metrics if the reviewer could kindly provide suggestions. As requested, we also report the FVD results below, where our method demonstrates superior performance compared to the baselines.
>
> | Method | TrajectoryCrafter | DaS    | Wan2.1-Inpainting | Ours       |
> | ------ | ----------------- | ----------------- | ----------------- | ---------- |
> | FVD↓   | 355.23      | 363.36           | 280.06           | **157.11** |
>
> ---
>
>
> ### Q2: Dataset release.
>
> While we are unable to include external links in the response due to the rebuttal policy, the dataset has been fully prepared. We will ensure it is released promptly upon paper acceptance and include the link in the final version.
>
> ---
>
> ### Q3: Did you retrain the baselines on your dataset?
>
> Thank you for the thoughtful question. We have attempted to retrain the baseline models on our dataset, but their performance deteriorates as training progresses. This is because prior works rely on input conditions and outputs that are temporally aligned, whereas our task involves predicting future observations with temporally misaligned inputs and outputs. The data pairs are constructed accordingly, making the problem more challenging. Without tailored architectural modifications, baseline models struggle to adapt. Moreover, since all baselines are pretrained on general video datasets such as OpenVid and MiraData, which share a similar domain with ours, the domain gap is minimal.
>
> ---
>
> ### Q5: Wall-clock time per generated 30-frame video and GPU memory footprint.
>
> Our memory footprint is approximately 25GB, and generation FPS is 1.8 minutes for 30 frames on an Nvidia A800.
>
> ---
>
> ### Q6: Performance degrades when the depth estimator produces noisy maps.
>
> We construct a new subset of 100 highly dynamic scenes covering both real-world and game-engine domains, including rainy and snowy weather as well as day and night environments from Sekai [1], a large-scale video world exploration dataset. Our evaluation reveal depth estimator perform well for night scenes and rainy/snowy environments, though the TSDF fusion stage may not fully filter out dynamic elements, leading to artifacts. However, our video generation backbone leverages static point cloud rendering as conditioning and integrates clean context into the video latent space. This design allows the self-attention layers to effectively capture cross-view dependencies, enabling the system to *repair distortions* while maintaining 3D consistency. As shown below, our results on this challenging subset remain competitive compared to other methods.
>
> | Method            | PSNR↑       | SSIM↑      | LPIPS↓     | FVD↓       | Aesthetic Quality | Imaging Quality | Temporal Flickering | Motion Smoothness | Subject Consistency | Background Consistency |
> | ----------------- | ----------- | ---------- | ---------- | ---------- | ----------------- | --------------- | ------------------- | ----------------- | ------------------- | ---------------------- |
> | TrajectoryCrafter | 9.8791      | 0.3497     | 0.5892     | 862.73     | 0.4846            | 0.5907          | _0.7158_              | 0.9866            | 0.9533              | 0.9425                 |
> | DaS               | _10.0074_     | _0.3849_     | _0.5719_     | 962.46     | 0.5102            | 0.6272          | 0.6421              | 0.9849            | 0.9565              | 0.9508                 |
> | Wan2.1-Inpainting | 10.0013     | 0.3811     | 0.5797     | _834.30_     | **0.5151**        | **0.6709**      | 0.6276              | _0.9893_            | **0.9583**          | **0.9668**             |
> | Ours              | **19.0984** | **0.6789** | **0.2228** | **256.18** | _0.5123_            | _0.6347_          | **0.8526**          | **0.9903**        | _0.9570_              | _0.9602_                 |
>
> (Bold is best, italic is the second.)
>
> [1] Li, Zhen, et al. "Sekai: A Video Dataset towards World Exploration." *arXiv preprint arXiv:2506.15675* (2025).

---

> > ### Comment · Reviewer_ddFV · 2025-08-04
> > **Comparisons with recent works.**
> >
> > Do you have any results Gen3C or additional baselines now?

---

> > > ### Author Response · Authors · 2025-08-05
> > > **Performance Comparison with GEN3C**
> > >
> > > Thank you for the thoughtful suggestions. We evaluate GEN3C on our testset. Specifically, we input our static point rendering and mask video as their rendered warp images and rendered warp masks into the model. As shown in the table below, we outperform GEN3C across all metrics. This is because GEN3C primarily focuses on 3D static NVS task and 4D NVS task in the same temporal sequence, whereas our work addresses the demand for temporally progressive NVS task. Additionally, we will include qualitative results in the final version for more intuitive comparisons, alongside the quantitative results.
> > >
> > >
> > > | Method | Aesthetic Quality | Imaging Quality | Temporal Flickering | Motion Smoothness | Subject Consistency | Background Consistency |
> > > |--------|-------------------|-----------------|---------------------|-------------------|---------------------|------------------------|
> > > | Gen3C  | 0.5203            | 0.5654          | 0.7179              | 0.9882            | 0.9178              | 0.9433                 |
> > > | Ours   | **0.5835**            | **0.6701**          | **0.7580**              | **0.9886**            | **0.9359**              | **0.9506**                 |

---

> > > > ### Comment · Reviewer_ddFV · 2025-08-05
> > > >
> > > > Thank you for the comparison, please update them in camera ready.

---

> > > > > ### Author Response · Authors · 2025-08-05
> > > > >
> > > > > We sincerely thank you for your constructive feedback and your engagement during the rebuttal stage. We will make sure to incorporate the additional comparisons into the final version.

---

### Official Review · Reviewer_arRQ · 2025-07-03

**Clarity:** 3
**Significance:** 2
**Originality:** 3
**Rating:** 5
**Confidence:** 3

**Summary:**

The paper introduces a geometry‑grounded memory framework for autoregressive video‑diffusion “world models” that fuses three complementary memory streams including short‑term working memory (recent latent frames), long‑term spatial memory (TSDF‑filtered global point cloud of static elements), and long‑term episodic memory (sparse key reference frames). It details algorithms for writing to and reading from these memories and integrates them into a DiT‑based latent diffusion backbone. Experiments show large gains in view‑recall consistency, VBench video quality, and human preference over leading geometry‑guided baselines. Extensive ablations confirm that each memory component contributes distinct benefits to long‑horizon coherence.

**Questions:**

1. Broader baseline set: please include experiments against recent long‑context compression/linear‑attention methods (e.g., LinGen, StreamingT2V, History‑Guided Video Diffusion). This would clarify whether geometric memory is strictly superior or complementary.
2. Dynamic‑scene analysis: construct a subset where more (e.g., >50%) pixels are dynamic (crowds, traffic). Report whether spatial memory degrades and, if so, propose possible mitigation solution.
3. Ablation on memory size: explore trade‑off between point‑cloud resolution / key‑frame buffer length and compute. Present memory foot‑print and generation FPS to justify practicality.

**Ethical Concerns:**

["NO or VERY MINOR ethics concerns only"]

**Final Justification:**

The authors addressed most of my concerns during the rebuttal phase. I believe the work is novel and provides valuable insights.

**Limitations:**

Yes

**Quality:**

2

**Strengths And Weaknesses:**

Strengths:
1. Technically sound. The memory design is well‑motivated, the TSDF fusion and key‑frame selection are precise, and ablations isolate each subsystem’s impact.
2. Experimental analysis. Quantitative, qualitative and human studies all align with claims, demonstrating state‑of‑the‑art long‑term consistency while maintaining video quality.
3. Significance. Bridging 3D perception with video diffusion offers a reusable path toward infinite‑length, scene‑consistent generation for graphics, robotics and simulation communities.

Weaknesses:
1. Baseline coverage is narrow—recent long‑context or state‑space video diffusion models are absent, leaving open whether geometric memory outperforms purely temporal approaches.
2. The method is trained only 6 k iterations on 8×A100 GPUs; scalability to larger backbones or diverse datasets is unverified.
3. The proprietary dataset is promised but not yet released, and spatial memory struggles in highly dynamic or drastic‑camera‑motion scenes (failure case, Figure 6).

---

> ### Author Rebuttal · Authors · 2025-07-31
>
> Thank you for your detailed review and insightful comments. We will address your questions and concerns below and in the revised paper.
>
> ---
>
> ### W1&Q1: Broader baseline set.
>
> Thank you for the thoughtful suggestions. As LinGen only has an unofficial implementation and does not provide a pretrained checkpoint, it is difficult to include in the current rebuttal phase. Instead, we conducted experiments on StreamingT2V and History-Guided Video Diffusion (Dfot, trained on RE10K) using the RE10K testset for comparison. The results are shown below:
>
> | Method        | Aesthetic Quality | Temporal Flickering | Motion Smoothness | Subject Consistency | Background Consistency |
> | ------------- | ----------------- | ------------------- | ----------------- | ------------------- | ---------------------- |
> | Dfot          | 0.4275            | 0.6800              | 0.9877            | 0.8630              | 0.9190                 |
> | Streaming_t2v | **0.4683**        | **0.8510**          | _0.9926_            | _0.8706_              | _0.9136_                 |
> | Ours          | _0.4630_            | _0.7838_              | **0.9936**        | **0.9583**          | **0.9576**             |
>
> We achieve second-best results in terms of aesthetic quality and temporal flickering, while significantly outperforming other methods on subject consistency and background consistency, thanks to the incorporation of geometric memory. These improvements are well aligned with the core motivation behind our model design.
>
> ---
>
> ### W2: Scalability to larger backbones or diverse datasets.
>
> Thank you for the thoughtful feedback. We conducted our experiments on CogVideoX and the current dataset due to limited computational resources, yet the results are already highly encouraging. We would like to clarify that scaling up from the 5B CogVideoX model to larger backbones, such as the 14B Wan2.1 model, is feasible, as they share similar DiT architectures and the parameter gap is not substantial. Furthermore, although our task focuses on camera-guided video generation, the training data pairs we use ultimately fall under the caption-video category, similar to those employed in I2V and T2V tasks. Therefore, scaling up the dataset is entirely reasonable. Due to the limited time for rebuttal, we plan to explore larger backbones and a more diverse dataset than the current version for the final version.
>
> ---
>
> ### W3: Dataset release.
>
> While we are unable to include external links in the response due to the rebuttal policy, the dataset has been fully prepared. We will ensure it is released promptly upon paper acceptance and include the link in the final version.
>
> ---
>
> ### W3&Q2: Dynamic‑scene analysis.
>
> Thanks for the insightful comment.
>
> **First**, to handle highly dynamic scenes with drastic camera motion, our strategy involves scaling the depth within the current range and reducing the voxel volume size. This adjustment yields a denser, filtered static point cloud, mitigating the issue of incomplete static-dynamic decoupling caused by extreme motion. By refining the static representation, we prevent excessive sparsity in the static spatial memory, thereby improving overall generation quality.
>
> **Second**, We construct a new subset of 100 highly dynamic scenes covering both real-world and game-engine domains, including rainy and snowy weather as well as day and night environments from Sekai [1], a large-scale video world exploration dataset. Specifically, for the reconstruction metrics, we far surpass other methods, which reflect our ability to keep 3D consistency and control camera accuracy. In terms of vbench metrics, we show superior video quality, although aesthetic quality and imaging quality lag slightly behind Wan2.1-inpainting, as mentioned in Sec.4.1 of our paper, this is due to their advanced backbone. Overall, our method still shows robust performance in these difficult cases.
>
> [1] Li, Zhen, et al. "Sekai: A Video Dataset towards World Exploration." *arXiv preprint arXiv:2506.15675* (2025).
>
> | Method            | PSNR↑       | SSIM↑      | LPIPS↓     | FVD↓       | Aesthetic Quality | Imaging Quality | Temporal Flickering | Motion Smoothness | Subject Consistency | Background Consistency |
> | ----------------- | ----------- | ---------- | ---------- | ---------- | ----------------- | --------------- | ------------------- | ----------------- | ------------------- | ---------------------- |
> | TrajectoryCrafter | 9.8791      | 0.3497     | 0.5892     | 862.73     | 0.4846            | 0.5907          | _0.7158_              | 0.9866            | 0.9533              | 0.9425                 |
> | DaS               | _10.0074_     | _0.3849_     | _0.5719_     | 962.46     | 0.5102            | 0.6272          | 0.6421              | 0.9849            | 0.9565              | 0.9508                 |
> | Wan2.1-Inpainting | 10.0013     | 0.3811     | 0.5797     | _834.30_     | **0.5151**        | **0.6709**      | 0.6276              | _0.9893_            | **0.9583**          | **0.9668**             |
> | Ours              | **19.0984** | **0.6789** | **0.2228** | **256.18** | _0.5123_            | _0.6347_          | **0.8526**          | **0.9903**        | _0.9570_              | _0.9602_                 |
>
> (Bold is best, italic is the second.)
>
> ---
>
> ### Q3: Ablation on memory size; Present memory foot‑print and generation FPS.
>
> 1) Since we use point cloud renderings as the conditioning input, the resolution of the point cloud does not directly impact the memory footprint. Additionally, the key-frame buffer is relatively small compared to the overall model size, so there is no significant trade-off within the practical range.
> 2) Our memory footprint is approximately 25GB, and generation FPS is 3.2 minutes for 49 frames on an Nvidia A800. We would like to clarify that the primary focus of our paper is on incorporating long-term and spatial memory into video models, and we do not claim real-time performance. However, we fully agree that bridging academic research and practical deployment is an important direction. In future work, we plan to explore lightweight alternatives and acceleration strategies such as Distribution Matching Distillation to further reduce memory usage and improve inference speed.

---

> > ### Comment · Reviewer_arRQ · 2025-08-05
> >
> > The authors adequately responded to my questions, and I have no other questions left. I appreciate the authors taking the time to provide more experimental results and analysis. Please include these discussions in the revised version.

---

> > > ### Author Response · Authors · 2025-08-05
> > >
> > > We sincerely thank you for your constructive feedback and engagement during the rebuttal stage. We will make sure to incorporate the additional experimental results and discussions into the revised version.

---

### Note · Authors · 2025-08-16

We sincerely appreciate the ACs, SACs, PCs, and all reviewers for their time and effort in the review process. We received positive ratings from all four reviewers and are deeply grateful for their detailed feedback and recognition of our work's key contributions. We are highly encouraged that: Reviewer arRQ found our work **technically sound, well‑motivated, experiments align all with claims, demonstrating state‑of‑the‑art long‑term consistency, and that bridging 3D perception with video diffusion offers a reusable path toward infinite‑length, scene‑consistent generation**; Reviewer ddFV described our work **poses a timely problem with straight forward implementation and clear writing style**; Reviewer Kmfd considered our work to benefit from **good formulation that involves spatial, working and episodic memory, results, and ablation experiments**; Reviewer e9qv described our work **can be useful building blocks/comparisons for future work**.

Based on their valuable and insightful comments, we have carefully updated our manuscript as follows:

- As requested by reviewers, we provide new experimental results, including:
  **1)** Comparisons with broader baselines (Streaming_t2v, Dfot, GEN3C), showing consistent improvements;
  **2)** A newly constructed subset with highly dynamic scenes covering both real-world and game-engine domains, where our method demonstrates robust performance and clear advantages over baselines;
  **3)** Quantitative view recall consistency results in the ablation study, demonstrating the effectiveness of our designs;
  **4)** FVD results, showing clear advantages over baselines.
- We further discuss scalability to larger backbones and diverse datasets, the potential of explicit and implicit 3D representations, the choice of 3D reconstruction methods, and provide more in-depth analysis of our approach.
- We clarify implementation details, experimental setups, metrics, GPU memory usage, and inference speed.
- We confirm that the dataset will be released promptly upon paper acceptance and include the download link in the final version.

During the rebuttal period, we were encouraged to receive further positive feedback and recognition from the reviewers. We will incorporate all revisions above into the final version. Once again, we thank the ACs, SACs, PCs, and all reviewers for their thoughtful contributions.

---

### Decision · Program_Chairs · 2025-09-17

**Decision:**

Accept (poster)

**Comment:**

This paper introduces a geometry-grounded long-term spatial memory for video world models, combining working, episodic, and spatial memory to improve long-horizon consistency. The approach is technically sound, clearly presented, and shows strong gains over recent baselines with thorough ablations and additional results provided in rebuttal.

Pros:
* Novel memory framework with clear motivation.
* Strong empirical results (quantitative, qualitative, human studies, FVD).
* Practical integration into diffusion backbones; useful for multiple domains.
* Rebuttal addressed reviewer concerns with extra baselines and dynamic-scene results.

Cons:
* Baseline coverage remains somewhat limited.
* Scalability to larger models/datasets not yet demonstrated.

Despite some open issues, the paper makes a clear and impactful contribution to long-term consistent video generation. Therefore, I'd recommend acceptance.